# Interconnections: A Systems History of Science, Technology, Leisure, and Fear

**Fred Phillips** [1,2] 

1   Anderson School of Management, University of New Mexico, Albuquerque, NM 87131, USA; phillipsf@unm.edu
2   School of Economics and Management, Tongji University, Shanghai 200092, China

**Abstract:** It is well known that technological change causes social change, and vice versa. Using system and historical perspectives, this article examines that truth at a finer level of specificity, namely, that social perceptions of interconnectedness influence the progress of science and technology, and that conversely, as 21st-century technology makes us in fact more connected, society's anxieties shift. From the science/technology side, we look at interdisciplinary research, system and complexity theory, quantum tech, and the Internet, exploring how these interact and cause changes in social attitudes—fears, conspiracy theories, political polarization, and even entertainment trends—some of which are surprising, and some dangerous. The article's systems view helps make sense of current environmental, political, and psychological crises. It combines original ideas with those of several prominent thinkers, to suggest constructive actions.

**Keywords:** systems; history of science and technology; science and society; terror

---

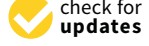

> *Terror is the normal state of any oral society, for in it everything affects everything all the time.*

> –Marshall McLuhan [1]

## 1. Introduction

Harold Linstone [2] pointed out that matters important to our lives had become simultaneously local and global. Phillips [3] added that these matters were also getting simultaneously larger (e.g., bigger companies) and smaller (nanotech). Revisiting these ideas some years later, Linstone and Phillips [4] wrote, "with the passage of time . . . these concerns come into sharper focus and prompt some further comments". The passage of another eight years prompts a still wider look, from a systems perspective, at how our social/psychological/technological world has got to be the way it is in 2021.

The present paper offers a brief recap of pertinent system concepts, leading to a conjecture about how science—which is a social endeavor—evolved toward the reductionist and back toward the holistic. It then focuses on our psycho-social response to science's rediscovery of our connectedness. Examples illustrate the systemic drivers of important cyclical trends and of trends toward polarization. Emphasis is given to interpretation and implications of these trends.

The paper's systems view leads to a conjecture that science, unwittingly mirroring society, revolutionized itself twice, shifting from holism to reductionism and back again. It argues that history shows no point attractors, but is more likely to be cyclic in important ways, and that terrorist acts are unavoidable in a hyper-connected society. Distinguishing among ameliorative measures that are systemic, technocratic, and merely radical, a list of recommended personal, social, and governmental actions concludes the paper.

## 2. A Brief Mention of System Concepts

In what follows, we will make reference to system theory. Rather than analyzing the behavior of individual entities, system theorists focus on the communication or other

interactions among entities [5,6]. The entities may be human, animal, or machine, and important subfields of system science address questions of human−machine communication and control [7,8].

A system is not just a set of entities and interactions ("nodes and links"). It is the set of generative rules governing the birth and death of nodes and links, and the changing intensity of connections [9]. That is to say, it is a matter of dynamics. This paper will focus on historical dynamics, with examples from tribal space, social space, environmental space, personal space, and cyberspace.

Other system concepts playing roles in this story are the butterfly effect, periodic attractors, holism vs. reductionism, variety, feedback, path-dependence and lock-in, and networks. Path-dependence simply means that history matters. Attractors are the conditions toward which a complex system may trend: Point attractors indicate stasis; periodic attractors mean cyclic behavior; and chaotic attractors have no easily described pattern and are to be avoided in most social and engineering systems.

## 3. Connectedness, from Hunter−Gatherers to Agrico-Religious Societies

McLuhan [1] implies that early human societies perceived every rock, tree, and animal as housing its own spirit or sprite, perhaps benevolent, but often malicious. Life's unpredictability, not to mention the effort of propitiating multiple magical beings, must have been psychologically wearing.

In the Middle East, sometime following the preliterate hunter−gatherer societies Marshall McLuhan addresses, there arose the Abrahamic religions. Their injunctions were: "Fear God. Fear nothing and no one else." This must have been a great relief to believers. Rejecting a diffuse fear of their total surroundings, adherents could focus their fears, and manage them through confession and via the religious institutions' other mechanisms.

Other features of these agricultural societies included the separation of labor, which introduced wealth inequality, and eventually led to a leisure class, a class insulated from many of life's dangers and uncertainties.

## 4. A Speculation on the Evolution of Science

When McLuhan's oral society gains enough leisure to develop a written language, "leisure" comes to mean not simply a few hours off work, but also some insulation from the terrors of the interconnected world. Enough insulation so that one could safely direct one's mind to matters other than immediate survival.

Obviously a number of human societies achieved this, with some—perhaps particularly in Europe in the second half of the last millennium—able to extend this secure leisure to a sizeable thinking class. Thorstein Veblen [10] characterized the leisure class as parasitic. Though this may have accurately described many of its members, its Newtons, Faradays, and Darwins turned personal funds or sponsorships into transformative scientific breakthroughs. This class of scholars and inventors were able to separate from the sphere of holism and develop the reductionist science that has held sway for some centuries.

Reductionism means that everything is *not* connected to everything—or at least that we may pretend that it is not. Instead, a natural phenomenon (dependent variable) is a function only of a few main causes (independent variables), maybe with a small number of mediating and moderating variables thrown in. Other forces at large in the world are said to have an influence "too small to matter," subsumed in the model's error term. Very often, the model encompasses one-way influence only, with no feedback effects.

These models could sometimes be tested in a laboratory (or in Darwin's case, an island), which further *insulated* the experiment from variation in non-treatment variables.

Therefore, following Thomas Kuhn [11], the philosopher of science who first held that science is a social endeavor and not a disembodied, "objective" activity, I offer this

speculation: *Reductionist science arose as a result of, and in probably unconscious imitation of, the growth of insulated social leisure.*[1]

## 5. System Science—The Pendulum Swings Back

Reductionist science remained king in the West—and I say "king" to highlight its gender (and racial) bias—until the pivotal events of the 1946–1953 Macy Conferences on cybernetics. These conferences spawned breakthroughs in system theory, cybernetics, cognitive science, and information theory (e.g., [7]).[2] They spurred university interdisciplinary programs. Why did this shift happen at this time? Largely because in contrast to World War I's, the tools of WWII were to a greater extent weapons of communication (radar, machine-based decryption, etc.) rather than weapons of force. These demanded a new science. Reductionism persists to this day, and indeed is needed, but many scientists have turned in a more holistic, systemic direction.

Now, it seemed everything was connected again, via the Internet and big data. Microsoft CEO Satya Nadella declared, "Everything is going to be connected to cloud and data"[3] Ericsson's CTO Erik Ekudden [12] sees

> a fully digitalized, automated and programmable world of connected humans, machines, things and places. All experiences and sensations will be transparent across the boundaries of physical and virtual realities . . . . Soon, there will be hundreds of billions of connected physical objects with embedded sensing, actuation and computing capabilities, which continuously generate informative data [12].

By 2020 connectedness reached a still newer extreme. It was not just our coding that connected us, nor even the rapid transmission of the Covid-19 virus: Connectedness turned out to be in the fabric of the universe itself. Quantum technology and quantum action at a distance re-affirmed that everything—that is to say, *everything in the universe*—is connected.

Marshall McLuhan might say we have come full circle.

## 6. How We Respond to Being "Reconnected"

*There is a natural progression of increased connectivity among humans. Groups of people start off simply sharing ideas, tools, creations, and then progress to cooperation, collaboration, and finally collectivism. At each step the amount of coordination increases.*

–Kevin Kelly [13]

McLuhan's statement about preliterate societies seems to apply also to our society today, in which the word "terror" appears in the news daily. Simply put, re-connectedness re-awakens terror. This section discusses how our science, industry, psychology, entertainment, and time allocation have adjusted to our renewed state of connectedness.

### 6.1. Science, Technology, and Industry

The late 20th century brought growing globalization of trade and investment, international transmission of epidemic disease, satellite communication, and ultimately (drum roll, please) the Internet. A well-known technology magazine is titled WIRED. This is to say that everything became interconnected again.

Science became more holistic. Outstanding examples include the work of Ilya Prigogine [14,15] on open and dissipative structures, and the work of the Santa Fé Institute on complexity theory [16], both of which acknowledge feedback (circular causation).

---

[1] Much of this European science grew from the Aristotelian tradition. Bertrand Russell wrote, "Athenian slave owners, for instance, employed part of their leisure in making a permanent contribution to civilization." Russell portrayed leisure in the context of anti-slavery and unjust economic inequality, however, rather than as an escape from terror. https://www.brainpickings.org/2018/12/27/in-praise-of-idleness-bertrand-russell/.

[2] Why did this shift happen at this time? Largely because in contrast to World War I's, the tools of WWII were to a greater extent weapons of communication (radar, machine-based decryption, etc.) rather than weapons of force. These demanded a new science.

[3] https://www.brainyquote.com/quotes/satya_nadella_597177.

In industry, we shifted from a capital economy to a knowledge economy, with enhanced feedback. Managers now must adjust to product and service reviews on social media. Where an early 20th-century firm's muscle workers networked only locally, exchanging gossip only in the neighborhood tavern and creating little new knowledge, today's knowledge workers' networking is worldwide. It "amplifies knowledge across distance, [and] creates new knowledge" [17].

### 6.2. Psychology

We have had to understand, or at least suffer from, the complexities that create climate change, and the grievances of people who turn to violence and cybercrime, knowing that their small actions can make havoc—this is the "butterfly effect"—across huge swaths of lives and property.

How do we feel about this? Mandrone [18] captures the psychological terror of reconnectedness:

> The digital society is complex because it faithfully represents reality. It is a more precise map, [a more finely detailed] representation of heterogeneity, which includes every identity. Reductionism has been for too long a tool to simplify reality, to make it easier to manage, but in this way, we are losing many nuances. The new role of information ... represents a new dimension for individuals, States, and companies. In these weeks of apprehension, there is a counter-intuitive relationship between understanding and fear, when insecurity should decrease as knowledge grows.

Popular culture tries to ease the terror of interconnectedness [19]. Disney's *Beauty and the Beast,* for example, with its dancing teapots and talking candlesticks, portrayed the sprites within inanimate objects as siding always with the good guys. This was timely, as newly marketed smart speakers and smart home appliances were both helping us and spying on us. Yet we also deal with fear by deliberately scaring ourselves with fairy tales (as was their original purpose). The evil witch in Disney's *Sleeping Beauty* had a magic mirror that allowed remote viewing and led to no good. Our real life, Ekudden [12] remarks, will "lead to the emergence of the Internet of Senses, which combines visual, audio, haptic and other technologies to allow human beings to have remote sensory experiences".

### 6.3. Mexicans, Monoliths, and Meteors

This soothing pop art did not work on everyone. Individuals allergic to complexity spawned or embraced conspiracy theories like Q-anon. Their delusions seem to be constructed as follows: "*My* life is simple, but the *other guy* is doing something complex, and *that's* what I'm afraid of".

Similarly, we look to the other, to aliens, both as threat (Trump's nonsensical tirades against Mexicans) and as possible saviors—viz., the mythologies arising around the desert monoliths [4] and the interstellar object Oumuamua [20].

### 6.4. Leisure

And what has become of leisure? Americans' median number of leisure hours dropped from 26 hours in 1973, to 20 hours in 2007 to 16 hours in 2008.[5] Thompson [21] tells us that (in a feedback effect) spending more time on work leads to getting accustomed to more work, and to identifying with it. Contrary to the stereotypically "idle rich," Thompson says, for this reason, rich people were working increased hours in 2016. At the other end of the

---

[4] https://www.cnet.com/news/mysterious-monoliths-in-utah-romania-and-california-everything-we-know-so-far/.

[5] https://apps.prsa.org/SearchResults/view/7722/105/Americans_today_have_less_free_time_study_says. These numbers include the ambiguous category of 'exercise'—which for some is truly fun, while for others it is an element of self-care. If one counts self-care exercise as leisure, one might as well count tooth brushing as leisure. The leisure-hours numbers are further clouded by today's habit of multi-tasking, e.g., listening to music while working. One may note the appeal of blogs with titles like "I spent a whole week off the grid."

income spectrum, leisure has become co-extensive with unemployment. It is unemployed youngsters lucky enough to live with their parents, who have leisure time on their hands.

The official definition of terrorism is the deliberate targeting of non-combatants. Yet we may also say that terrorism is in the mind of the terrorized.[6] Terror has returned, not just due to evil deeds of some non-state actors, but because we are aware that everything is once again connected to everything.

### 7. The Pendulum Keeps on Swinging: Examples

*In complex systems, when you build in more safeguards and redundancies, you increase the probability of error.*

–Charles Perrow [22]

Thus, history shows beats of connectedness and isolation—cycles of tolerance, desire, and opportunity for connectedness, and cycles of tolerance, desire, and opportunity for isolation. This section offers more examples of socio-technical system dynamics leading to cycles and to polarization.

### 7.1. Cycles

Society is cyclic, though each turn may be driven by a different reason. Disease, cities, new technologies, and globalization provide more examples of cyclicity.

#### 7.1.1. Disease

My parents, born in the 1920s before penicillin, were wary of "germs". My father had done business in postwar Japan, and when I later studied in that country, he recommended that I visit a bathhouse where attendants would bathe and massage customers with towels and sponges, a relaxing experience "untouched by human hands". My generation, accustomed to antibiotics, believed that it is very nice to be touched by human hands! In the Covid-19 era, we are again learning "social distancing". We have swung from "high tech, high touch" [23] to an incipient "low-touch economy".

#### 7.1.2. Cities

Until the 1940s or so in the USA and the UK, gentry lived in the countryside. Perhaps they kept a flat in the city for business and entertainment purposes, but generally they regarded cities as pits of crime and sin. By the 1950s and 60s, city residency had become fashionable. The era featured much urbanization and a growing urban middle class, though racial redlining made for a less than rosy total picture.

By the 1970s, US cities had become unpleasant, and, some thought, ungovernable. Young people yearned for the countryside. Joni Mitchell sang, "Got to get ourselves back to the garden" [of Eden]. It was many years later before Caucasians of my generation understood this phenomenon's racist undercurrent. In any case, in the early 2000s Generation Z and the millennials returned to cities. The financial businesses had never left, and now it was speedy Internet, available only in the cities, that enabled that industry's fast trading of securities.

In 2020, Covid-19 drove a new urban exodus—although ever-stronger hurricanes had already thinned out the populations of U.S. southeastern coastal cities.

#### 7.1.3. Democratizing Technologies

Occasional "democratizing technologies" enable formation of skilled small businesses, until investors with more capital than skill—popularly called "roll-up artists"—corporatize them. The plow enabled family farms to thrive for centuries, until modern corporate farming. The sewing machine enabled small tailoring and dressmaking businesses, but then . . . sweatshops. Likewise, the personal computer gave rise to a greater number of

---

6  In another feedback loop, publicized instances of gun violence feed people's fear, and gun sales increase. www.science20.com/news_staff/gun_shootings_lead_to_more_gun_purchases-237389.

profitable small bookkeeping and tax accounting businesses, but then call centers were invented. There are now artificial intelligence packages for small businesses [24]; however, Harari [25] warns that A.I. can (will, he believes) lead to a total surveillance state, as large companies—and governments—know what you are going to do before you do it.

### 7.1.4. Globalization

Globalization came and went on longer cycles. A thousand years ago, a globalized Arab world, centered in Cairo, stretched from Morocco to Baghdad. Financial transactions and human travel flowed smoothly and safely across this huge geographic span. The 20th century saw a resurgence of globalization, driven by Western countries, though treaties favored international transfers of funds rather than flows of people.

China, once surrounded by tributary states and accomplished in naval exploration and trade, retreated within itself, and only now is reaching out again, with its Belt and Road initiative and other efforts to exert a global role.

However, China still tightly controls human and financial inflows and outflows. More generally, the 21st century has brought us a new rise of nationalist autocrats in many countries, and the renunciation of international treaties.

Financial globalization is again in retreat in the West, even as cultural globalization is on an upswing worldwide. The latter is driven partly by social media and movies, and partly by migrants [26].

### 7.1.5. Political Philosophies

US elections produce swings between dominance of one or the other political party. The parties represent, broadly, individualism (minimal government) versus collectivism. Like any philosophies, either one is dangerous—and neither is defensible—in its extreme version. Collectivism leads either to lack of coordination, or as in the Soviet Union, authoritarian uncoordination. Individualism ignores our interdependence. If one is only to take care of oneself and we are not taking care of each other, then the word "nation" has no meaning. The logic of individualism falls apart completely at the level of the family; it appears to be an excuse for small-scale male authoritarianism.

### 7.2. Polarization

The individualism−collectivism dichotomy leads to a discussion of other instances of polarization. In every case, as is well known, social media "echo chambers" lead vulnerable people to embrace ever more extreme views at either pole. This is a systemic feedback effect.

Inequalities in income and wealth have left many US families "one medical emergency away from poverty", even as the super-rich class is further enriched. Figure 1 shows the systemic connections among environmental problems, authoritarian government, inequality, and attitudes toward science. The plus signs (+) in the Figure indicate that the more that happens at the arrow's tail, the more will happen at the arrow's head. The entire picture demonstrates a vicious cycle.

Families suffering from environmental hardships—drought, flooding, etc.—seek a savior from any quarter. The autocrat identifies culprits, usually foreigners and usually at random, to blame for the catastrophe and promises to chase them down. The autocrat has no intention of spending money on the problem; on the contrary, he courts and favors the wealthy, in order to shore up his own status. This is crony plutocracy, and it exacerbates economic inequality. At the same time, the autocrat and his plutocrats dismiss scientific objections to his policies. He defunds science, with the result that fewer scientific resources are available to ameliorate environmental problems. These problems get worse, and the cycle repeats. Each time around generates more fans of the autocrat's persuasive but casuist storytelling, and at the same time generates more skeptics who gravitate to the opposition.

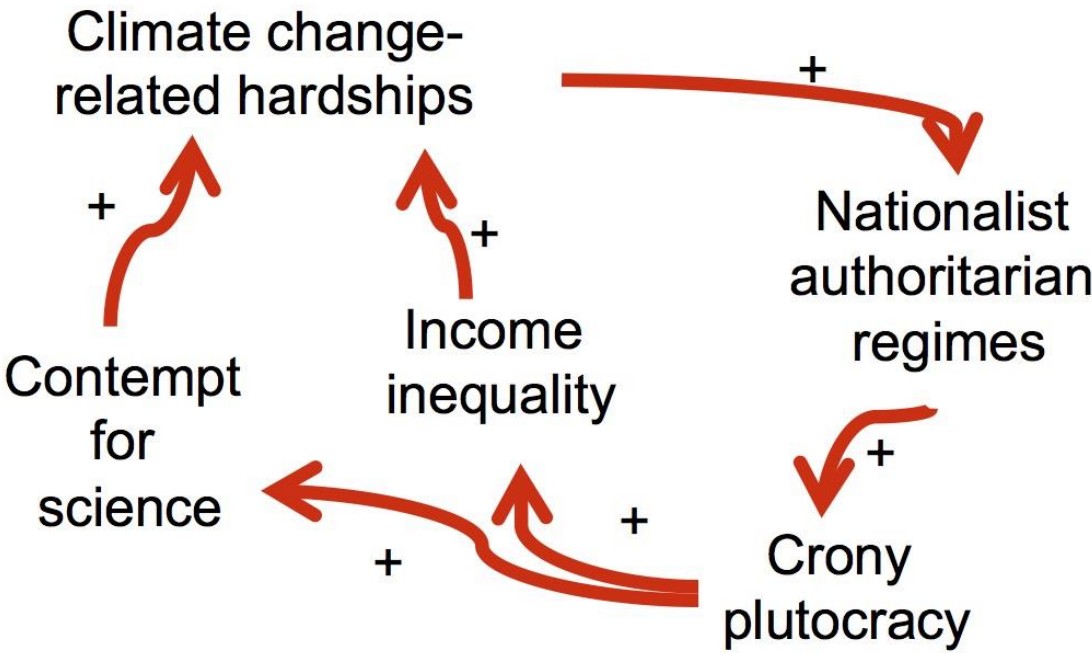

**Figure 1.** Influence diagram linking politics, economics, and science.

The cycle cannot continue forever. It creates what Turchin and Korotayev [27] call "pressure" that leads to social unrest. Those authors' "structural demographic theory" maintains that unrest erupts in response to a "trigger event". In their view, the 2020 death of George Floyd at the hands of Minneapolis police was such a trigger. Other events like extreme environmental hardship, or too many aspiring but failing to join the exclusive plutocracy, could trigger the breakdown of Figure 1's vicious cycle.

Mims [28] mentions more polarizing forces:

- As professionals work at home to avoid infection, "the pandemic may have permanently reduced how often people work from an office", with all that implies for real estate prices, urban non-professional jobs, and city tax receipts.
- A K-shaped recovery will keep professionals at work while eliminating the jobs of lower-paid people who, in the Covid-19 crisis, have truly proven (as Veblen predicted) to be "essential workers". The latter are store clerks, janitors, laborers, and office assistants.
- Stock prices are looking good even as unemployment skyrockets and GNP growth slows.
- Crisis-spurred technological innovations create a learning hurdle the less educated cannot meet. The digital divide widens.
- Those suffering most will be women and minorities.

Mims adds that Covid-19, compounding the current economic recession, will cause many small businesses to be wiped out. In the six months following March, 2020, [28], increasing the influence of the largest companies in each of those sectors. Walsh [29] further explains that even within those large companies, algorithmic management will eliminate much upward job mobility, making income inequality even more extreme.

The classic curve relating inflation to unemployment has unraveled again in 2020, as it did in the "stagflation" of the 1980s. Some may view this as a failure of a traditional correlation. A system scientist will view it as a phase transition, the causes and consequences of which are yet to be fully traced.

### 7.3. Working against Each Other

We face complexities of the connected and unpredictable universe, and we face complexities that we have brought on ourselves. (Linstone and Phillips [4] warned against

allowing the latter kind to get out of hand). Satellite launches (for 5G wireless and other purposes) offer a current example of how we may create a level of complexity that will destroy us.

The Environmental Health Trust [30] is pressing the US Federal Communications Commission to cease giving licenses to unlimited numbers of communication satellite launches.[7] The FCC may be doing this in an unauthorized fashion, but the legal issues are beyond the scope of this paper. What is relevant here is that:

- Cascades of space junk (fragments of damaged satellites) could bring down all global communications. Thousands more satellites are slated to be launched to support 5G and other communication bands, dangerously raising chances of disaster, the so-called Kessler Syndrome.[8]
- Meteorologists claim heavy wireless traffic in 5G bands will set back weather forecasting to 1980. The resulting failure to forecast the path of ever-stronger storms [31] could cost thousands, maybe millions of human lives [32].
- Astronomers fear the cloud of orbital objects resulting from massive launches of 5G satellites will hinder their ability to conduct science—and to track possibly dangerous asteroids [33].

Anxious not to cause further polarization, the Environmental Health Trust wishes to engage—not confront—the FCC with these difficulties, and come to a middle ground in which Internet may be delivered to a limited extent by satellite, the remainder to be delivered via the existing and proven infrastructure of optic fiber.

## 8. Implications

*Our sense of self, which has been evolving through our entire life, becomes larger and more inclusive as we mature. This expansion has been characterized . . . as moving from egocentric, to ethnocentric, worldcentric and finally 'kosmocentric' (i.e., embracing the entire subjective and objective universe).*

–Ginny Whitelaw Roshi [34]

These cycles arise from the attractors in a complex techno-historical system. History seems to have no point attractors. The gradual unification of mankind forecast by Reinhold Niebuhr in 1949 never happened [4]. The "end of history" forecast by Francis Fukuyama [35] never happened. History brings us back to similar places, for different reasons.

Extreme connectedness leads to polarization, via feedback. Mild polarization invites compromise. Extreme polarization sometimes leads to violent conflict, as happened to the USA in its Civil War. Thus Fareed Zakaria [36] remarks that despite revealing how interconnected we all are, the effect of Covid-19 "has been to make us more narrow, more nationalistic, more selfish".

The paragraphs below draw on the wisdom of additional prominent thinkers, in order to draw the implications of the resurgence of connectedness and terror. They urge personal, institutional, and social action to balance the universal with the local.

### 8.1. Havel

Vaclav Havel, the playwright who became the first post-Soviet president of the Czech Republic, perceives that global technological civilization, though here to stay, is only a "thin veneer" over an unchanged human nature, over an "immense variety of cultures, of peoples, of religious worlds, of historical traditions and historically formed attitudes" [37]. Havel goes on to note that:

even as the veneer of world civilization expands . . . ancient traditions are reviving, different religions and cultures are awakening to new ways of being,

---

7　https://ehtrust.org/eht-takes-the-fcc-to-court/.

8　https://www.spacelegalissues.com/space-law-the-kessler-syndrome/.

> seeking new room to exist . . . and to be granted a right to life . . . [and] a political expression.

Vaclav Havel sees this as a central challenge to every part of today's world,

> to start understanding itself as a multicultural and multipolar civilization, whose meaning lies not in undermining the individuality of different spheres of culture . . . but in allowing them to be more completely themselves.

> This will only be possible, even conceivable, if we all accept a basic code of mutual co-existence . . . one that will enable us to go on living side by side . . . .

> Yet such a code won't stand a chance if it is merely the product of the few who then proceed to force it on the rest. It must be an expression of the authentic will of everyone, growing out of . . . our original spiritual and moral substance, which [in turn] grew out of the same essential experience of humanity [37].

Havel asked whether his idea of a new, common creed was "hopelessly utopian". As students of management, we know that new products—and new ideas and memes—penetrate the market because there are "innovators" and "early adopters". Thus, we may reply to Havel, it *is* utopian to expect everyone to accept a code of mutual co-existence all at once. At first, and at any stage, some people will, and some people won't. I urge readers of this article to be innovators in seeking the common spirit that will unite us.

### 8.2. McLuhan

Marshall McLuhan's vision implies that terrorist bombers are somehow inevitable in a connected world—an integral expression of the nature of this world, rather than an external threat to it. You are reading the present essay because you have chosen to tread the path of positive, constructive action. You will be leaders and teachers, and you will remind those who look up to you that each day, each of us decides as individuals whether to be part of the problem or part of the solution.

### 8.3. Arendt

Hannah Arendt [38] wrote, "The ideal subject of totalitarian rule is not the convinced Nazi . . . but people for whom the . . . distinction between true and false . . . no longer exist".

Arendt goes on to identify loneliness as "the common ground for terror" and asserts that loneliness is both a weapon and a consequence of autocracy. Moreover, she says, the preparation for dictatorship is complete when people have lost contact with their fellows. She sees thought and logic as collective: Without them, people "lose the capacity of both experience and thought".

Arendt helps us see the irony of Covid-19. Even as we remain connected via Zoom and email, we feel the loneliness of staying home all the time. It seems commonly thought that citizens who wear masks are collectivists, concerned with each other's' health. Yet, loneliness might override. As SUNY Stony Brook president Maurie McInnis remarked to his students, "Some of you have shared that you feel isolated, lonely, and disconnected". Lonely maskers could be vulnerable to propaganda, conspiracy theories, and conversion to fascism.

### 8.4. Applebaum

Anne Applebaum, who writes brilliantly in *The Atlantic* on autocracy and nationalism, opined in 2018 that democracies eventually die for good. By the end of her essay, Ms. Applebaum [39] had reversed course, veering to the view presented in this paper, namely that democracies come and go in cyclic fashion. Really, this should have been evident from the period 1976–2020 in the USA, as the Democratic party and the ever more autocratic Republican party alternately captured the White House, and even earlier: Putnam and Garrett [40] noted the swings from the 19th-century Gilded Age, to progressivism and the New Deal, to the turbulent 1960s, then back to elitism in the Reagan era and forward.

## 9. Actions

> *In order to survive on this planet, we need to respond as one system, we need to come together as one human race and include all voices to navigate the dark night of racism, loss of biodiversity, climate change and war. We are called to work on ourselves, on our societies and on our relationship to our beautiful and fragile Mother Earth.*
>
> Patrick Cassidy[9]

What, specifically, can we do to realize Vaclav Havel's vision? An American business education magazine offered four answers to this question. These answers are:

- Social entrepreneurship
- Post-conflict planning studies
- Affirmative inquiry
- Personal relationships

To freely interpret each of these:

*Social entrepreneurship.* The best-known instance of this phenomenon is micro-loan programs, which provide both an ROI for the investor and sustenance for the needy. Even beyond those worthy results, however, these programs show young people that there are paths out of poverty that do not involve the taking up of arms.

*Post-conflict planning studies.* These efforts allow participants to form a positive vision in which their regional conflict is not unending, and in which desirable things will happen after the coming of the peace.

*Affirmative inquiry.* Guided discussion that focuses on the positive changes that can be made, rather than on blame and the negative aspects of current realities. Affirmative inquiry also involves building a classroom environment in which world political conflicts are not assumed to be "ongoing," and in which students feel empowered to ask, "How can I make a difference"?

*Personal relationships.* Classroom projects, summer institutes, and entire new schools are built around joint efforts of students from both sides of a conflict. Worthy social entrepreneurship projects are often the result.

I will add that at the management school where I work, we insist on mutual courtesy and respect among all the ethnicities and genders represented at the school. This is not because we naively believe that commerce and trade are the common factors that will create peace in the world. Rather, we believe along with Vaclav Havel that displaying respect in this way is simply right—a central part of the code of co-existence that will allow us to survive and thrive.

Resilience to Covid does not mean returning to the *status quo ante bellum*. Noting that great innovations in urban policy result from disasters—London's 1832 cholera epidemic, Chicago's 1871 fire, New York's blizzard of 1888—Thompson [41] suggests cities may benefit from these measures in the era of Covid:

- *Universal health care.* Covid-19 proved that when you get sick, everyone gets sick.
- *Clamping down on automotive traffic.* Pollution from internal combustion engines kills 50,000 Americans each year.
- *Turning vacant buildings into low-cost housing.* Reduces car commuting by allowing more families to live near downtown.
- *Updating ventilation standards.* Buildings with un-openable windows and poor HVAC are "perfect petri dishes" for the spread of disease.

In a memorable sentence, Thompson indicates the importance of systemic actions: "*New York did not react to the 1888 blizzard by stockpiling snow shovels.* It created an entire infrastructure of subterranean power and transit that made the city cleaner, more equitable, and more efficient". Though the press bruits claims that the post-COVID era will require radical restructuring of the economy—the World Economic Forum calls one such "the great

---

[9]　In a 2 November 2020 Facebook post, https://www.facebook.com/patrick.cassidy.9279/posts/10158653343097778.

re-set".[10] Mazzucato [42] is skeptical that we will have learned the lesson of 1888 New York. "Radical" is not necessarily "systemic".

Yong [43] lists nine fallacies in America's pandemic response, which I relate verbatim:

- "Serial monogamy" with particular solutions, rather than embracing several at once in wholistic fashion;
- "False dichotomies", for instance between sick and well, or between saving lives and saving the economy;
- The "theatricality" of preventive measures, their efficacy notwithstanding;
- "Personal blame" of those who spread Covid-19 and fall ill from it, above the pursuit of "systemic fixes";
- A desire to return to normalcy;
- "Magical thinking" about a fast end to the crisis;
- The "complacency of inexperience", as even doctors in yet-to-be hit areas "seemed to forget that viruses spread", unsold on the problem's severity given that they had not yet seen it where they lived;
- A "reactive rut" that lacked forward thinking;
- And the "habituation of horror", as the emergency has become the normal.

Yong is implying, of course, that we must avoid these fallacies. Most of them are failures of critical thinking or system thinking.

Linstone and Phillips [4] wrote, "Not surprisingly decision-making is becoming ever more problematic . . . The world is doing better than pessimists expected, but current decision-making structures are not producing good decisions fast enough and on the scale necessary to address global challenges". Lest this tempt anyone to favor autocracy, thinking that a dictator could make faster decisions, remember that with the exception of China, democratic nations dealt with the Covid-19 pandemic much better than the autocracies did [44].[11]

Linstone and Phillips [4] concluded, "It is essential that we strengthen our crisis management capability. In particular, training is needed at all levels . . . to make sound decisions rapidly when faced with unexpected threats, human and natural. In a large scale crisis, the decision process must cut across the normal organizational, governmental, and geographical boundaries". This might best be done within coordinating agencies, perhaps on the model of the US National Security Council, which Freeman and Rossi [45] cite as an exemplary cross-cutting agency in the intelligence arena.

In short, the experts cited in this concluding section advise that we take charge of the balance between the connectedness and isolation in our lives and our communities. Balance is key. The cosmic view advocated by the above-quoted Zen teacher G. Whitelaw is attractive, yet Buddhism is criticized for not specifying a role for the family [46]. One may be excused for wanting to put family first, even while striving for a wider vision. Balance is truly difficult.

Yet, remark Linstone and Phillips [4], "in a democracy, popular audiences are the core of the polity, and must apprehend the systemic principles if the polity is to be a learning organization". Lakoff [47] makes a good case that the ability to think systemically is formed by one's early upbringing and solidified at that time. He implies that it is not possible to convert linear thinkers into system thinkers. As authoritarians de-fund education—so that the next generation will not be able to think at all—the young will believe what they are told and will be unable to recapture economic and political power from the elite.

Early in the pandemic, New York Governor Cuomo imposed behavioral restrictions on New York City, but not on the more rural "upstate" New York. This kind of *adaptive policy* [48] recognizes that conditions differ across time and geography. Adaptive policies ease polarization, because persons at each pole are afraid that if the other comes into power, it will impose its will on everyone. "One size fits all" policies are not always necessary.

---

10  https://www.weforum.org/great-reset/.

11  I wrote this on the eve of the 2020 USA election, a moment in history when classifying the USA as a democracy was questionable.

When adaptive policies are needed, they should be made explicitly adaptive, at the time of drafting into law, regulation, or ordinance.

Havel noted that at one level, we are one common humanity, but that in another sense, we are culturally "multi-polar". This importantly implies that globalization should not be allowed to homogenize us: As Ashby [7] pointed out, when catastrophes introduce new variety into our environment, it is only variety in human responses that will let our species survive. Fortunately, much human variety and diversity remain in today's world. Havel's statement is also important because "multi-polar" can be more survivable than the bipolar situations described earlier in this paper. Indeed, this principle was the motive for structuring the US government in three branches, rather than two.[12]

## 10. Concluding Remarks

Writers have addressed these issues from disciplinary perspectives. For example, Finkel et al. [49] link the psychology of political polarization to "aversion, othering, and moralization". Psychologists at Decision Resources Inc. [50] note that the seeds of polarization are "volatility, suspicion, and oversimplification". The historian Arthur M. Schlesinger [51] saw America driven by cycles of idealism and pragmatism. Such works complement the present paper's systems view, which presents the same psychology as a function of feedback and fear of extreme connectedness. This paper applied systems principles to elucidate the interactions and cyclic behavior of society's larger problems—climate, urbanization, authoritarianism, and others. Particular attention was aimed at long cycles of connectedness, disconnectedness, and reconnectedness—and the psychological terror, coping mechanisms, and scientific trends accompanying these cycles.

The paper tied the rise of reductionist science to a social trend toward leisure and insulation from extreme connectedness, and the resurgence of holistic science to new discoveries and inventions of a "connectedness" nature.

It noted that the dynamics of our sociotechnical system lead us back to similar places (but for different reasons), with history never reaching a point attractor. Just as we return to our childhood home to visit parents, later to celebrate marriages and births, and again to dispose of the house upon the passing of our parents, each visit at a different stage of our life cycle, we return to the periodic attractors of the sociotechnical system. We can speculate (and hope) that these returns correspond to Whitelaw's stages of our consciousness or to Kelly's stages of our growing unity as a species—and that we will ever avoid the "point attractor" of our extinction.

Science fiction authors posit multiple alternate futures. The truth is different: We are living in multiple futures, all at once! The paper showed that seemingly inconsistent trends, e.g., disappearing financial globalization and growing cultural globalization, reductionist and holistic science, and electronic connectedness with COVID-driven distancing, continue to operate simultaneously, as Linstone and Phillips [4] noticed. Radical remedies may be needed for the resulting crises, but "radical" and "systemic" are not co-extensive: Radical without systemic may make problems worse. By the same token, systemic does not mean technocratic: The world's multiplicity, and the multiplicity of human constituencies, means authoritarian imposition of mathematical "solutions" will not suffice, either. Softer systemic measures are called for, as described in the sections above.

The paper drew on the wisdom of several writers in order to outline a constructive path to the future. Readers who are not averse to multiplicity, complexity, and systems thinking [52,53] will appreciate the dynamics of connection/disconnection and comfort/terror. They will see how these dynamics suggest policy and personal pinch points that can forestall or ameliorate crisis situations.

---

[12] https://www.science20.com/machines_organizations_and_us_sociotechnical_systems/the_magic_number_3_maybe_not-175095. We must, unscientifically, call it miraculous that the bipolar USA-USSR Cold War standoff did not lead to a third hot world war.

**Funding:** This paper was presented as a keynote speech of SOItmC 2020, and the publishing fee was supported by SOItmC.

**Conflicts of Interest:** The author declares no conflict of interest.

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
