# Peer review of "Interconnections: A Systems History of Science, Technology, Leisure, and Fear"

_2199-8531, doi:10.3390/joitmc7010014_

Round 1

Reviewer 1 Report

Comments to author(s):            

First of all, I sincerely appreciate the opportunity to read and comment on interesting topics. This paper intended to explain the interaction and cyclical behavior of various social problems such as globalization, polarization and authoritarianism, etc. from a systematic perspective. The author(s) focused on “long cycles of connectedness, disconnectedness, and reconnectedness – and the psychological terror, coping mechanisms, and scientific trends accompanying these cycles”.

This paper is judged to be excellent. It is expected to provide readers with an opportunity to think about the cycle of technological advances and changes in the social environment. It would be more meaningful to add professional opinions in the following areas:

  1. I think it would be all the more meaningful if this paper gave a detailed account of the current situation of Cobid-19 as well as the social phenomena that may occur after the end of Cobid-19. In particular, I would like you to explain not only the polarization within the country but also the phenomenon that occurs between countries all over the world.
  2. The explanation for leisure is that the rich will work more, and I wonder if the fourth industrial revolution will keep the rich and the unemployed the same phenomenon.
  3. In Democratizing technologies, There is a warning about AI that Harari (2018) said, and I would like to hear more from the author(s)’ opinion about its impact.

Thank you again for the opportunity to read this paper.

Author Response

Thank you for these very kind comments.

Your suggestion is, again, flattering, but I don’t think it’s wise to make an already long paper even longer. However,

1. This is a hard question. I’ve now mentioned, in a footnote the US-USSR cold war face-off, and added a reference on polarization. I’m not prepared to tackle the “between countries” question in more depth now. As for COVID, I have added a reference to Mazzucato’s article in Foreign Affairs. It’s a good clear article, that you may want to read.

2. Robots and A.I.s will take many jobs from the working classes, and provide services to those classes. The very rich classes will continue to demand and employ, humans. This is my prediction, but it has little to do with the present paper.

3. Harari’s prediction fits my scheme, laid out in the paper, that inexpensive technologies are first democratizing, then rolled up for exploitative purposes. Harari is a historian. Each of his readers must judge his grasp of technological details.

Reviewer 2 Report

The manuscript focuses on an interesting and relevant topic to the current academic debate, policy and practice. 

1) The main issue with the manuscript is that it reads like an opinion paper rather than an academic research paper. I suggest the paper type to be changes to comment, commentary, viewpoint or perspective. 

In the case the above is done the paper has potential to be considered for publication given the following changes are also done. 

2) At the abstract (and also at the introduction) clearly state what is the premise (aim) of the paper is and what insights of findings it generates.

3) Avoid having main sections that are only one paragraph. Instead either expand these sections to a page or remove the section header. 

4) The literature background of the manuscript is limited and needs to be expanded. 

5) Change the Section 10 summation heading to Concluding remarks. Plus there is no need for 10 sections best to use the traditional 1. Introduction, 2. Literature background, 3. Methodology, 4. Results, 5. Discussion, 6 Conclusion sections.

6) The manuscript also needs a careful language editing. 

Author Response

The manuscript focuses on an interesting and relevant topic to the current academic debate, policy, and practice. 

>> Thank you for this comment.

1) The main issue with the manuscript is that it reads like an opinion paper rather than an academic research paper. I suggest the paper type to be changes to comment, commentary, viewpoint or perspective. 

>> I have no objection to the editor changing the article type designation, as he may judge appropriate.

In the case the above is done the paper has the potential to be considered for publication given the following changes are also done. 

2) At the abstract (and also at the introduction) clearly state what is the premise (aim) of the paper is and what insights of findings it generates.

>> Thanks for this reminder, which I always need. I’m better at writing the body of papers than at writing intros. The introduction is now expanded.

3) Avoid having main sections that are only one paragraph. Instead either expand these sections to a page or remove the section header. Plus there is no need for 10 sections best to use the traditional 1. Introduction, 2. Literature background, 3. Methodology, 4. Results, 5. Discussion, 6 Conclusion sections.

>> Respectfully disagreeing. Each paper needs to be structured in a way that best presents its argument. As you’ve noted, this is not a regular research paper, so it need not be sectioned in the ‘traditional’ way.

4) The literature background of the manuscript is limited and needs to be expanded. 

>> I have added several references, particularly in the section explaining system theory, and in the “Actions” section. Also a ref on the Oumuamua space object, and a reference to Veblen’s Theory of the Leisure Class. Not mentioning that one was a terrible oversight in the originally submitted version of this paper.

5) Change the Section 10 summation heading to Concluding remark

>> Done.

6) The manuscript also needs a careful language editing. 

>> I have reviewed the language and corrected several typos and awkward wordings.

Round 2

Reviewer 2 Report

It can be accepted for publication following the paper type is changed to 'Viewpoint'.